# A data-driven approach for examining the demand for relaxation games on Steam during the COVID-19 pandemic

**Maximilian Croissant**[ID]⊕*, **Madeleine Frister**[ID]⊕

Department of Computer Science, University of York, York, North Yorkshire, United Kingdom

⊕ These authors contributed equally to this work.
* mc2230@york.ac.uk

**Data Availability Statement:** The data underlying the results presented in the study are available from SteamDB. URL: https://steamdb.info/ Github: https://github.com/SteamDatabase.

## Abstract

The COVID-19 pandemic has been a major source of stress for a majority of people that might have negative long-term effects on mental health and well-being. In recent years, video games and their potential positive effects on stress relief have been researched and "relaxation" has been an important keyword in marketing a certain kind of video game. In a quasi-experimental design, this study investigated the increase of average daily player peak (ADPPs) for the COVID period compared to the pre-COVID period and if this increase was significantly larger for relaxing games in contrast to non-relaxing games. Results showed a medium-sized increase of ADPPs over all types of games but no difference between relaxing games and non-relaxing games. These results are discussed in regards to their potential of presenting gaps between the current theoretical models of the influence of video games on mental health and actual observed player behaviour.

## Introduction

With an infection rate of over 60 million cases to date [1], the COVID-19 pandemic has reached the dimensions of a global health crisis, affecting almost every nation in the world. Besides the direct physical health consequences, which most commonly involve fever, coughs, and breathlessness, the pandemic negatively impacts mental health and psychological wellbeing to a not negligible extent. This does not necessarily include a direct infection with the virus but may rather be due to overall social and personal difficulties arising from the crisis, such as financial losses, uncertainty about the future, and public health measures like social distancing and contact restrictions. Common mental health responses to the COVID-19 pandemic are symptoms of anxiety, depression, and stress [2–4], with the latter being amongst the most frequently reported adverse consequences in the global population. For instance, Salari et al. [3] identified a stress prevalence of 29.6% in the general population as a reaction to the pandemic, based on a meta-analysis of 5 studies with a total sample size of over 9,000 individuals from various countries. High rates of post-traumatic stress disorder (PTSD), psychological distress, and stress during the COVID-19 pandemic have also been identified in a systematic review by

**Funding:** The author(s) received no specific funding for this work.

Xiong et al. [4], who incorporated data from several regions in Asia, Europe, and America in their analyses.

## Stress, mental health, and wellbeing

Stress has long been known to have a wider negative impact on mental health. Since the 1970's, there have been extensive reviews about the consequences of stress-inducing life events on psychological health, describing various risks such as developing depressive symptoms, anxiety disorders, or even schizophrenia [5–9]. Stress has not only been shown to have a negative impact on mental health, but also on behavioural variables, such as drinking, smoking, and violent behaviour [10–13]—variables that in turn adversely influence health and wellbeing. Greenberg et al. [14] further collected literature describing the emotional and cognitive changes that resulted of stress, giving more context to the dangers of stressful life events for mental health. Besides that, plenty of studies have reported an inverse relationship between stress and subjective life satisfaction, providing evidence of the unfavourable influences that stress has on psychological wellbeing [15–18].

## Video games as stress relief

These various and severe adverse health consequences demonstrate the importance of ways to reduce psychological distress. One promising method that gained popularity in recent years are video games. More and more studies provide arguments that link the role of video games to mental health improvements. As an example, casual video games (e.g., pinball, card games) have been shown to reduce physical stress responses, and with that psychological consequences of stress [19]. Similarly, Holmes et al. [20, 21] and Iyadurai et al. [22] noted the potential of Tetris to reduce flashbacks after trauma exposure in PTSD patients, showing that even pathological stress responses can be mitigated by playing video games. Systematic reviews on the topic further underpin the potential of video games to improve general mental health outcomes, including stress relief [23–26].

Indeed, there is an increasing interest for video games that go beyond the mere purpose of being entertaining—commonly referred to as serious games. More and more health practitioners rely on the health and educational benefits of such games, validating their therapeutic incremental value [27]. More importantly, there is evidence that people systematically play video games for recovery purposes or after straining situations [28], with middle-aged adults even citing stress relief as a main reason to play [29]. These findings can be extended to a wider group of participants: Ferguson and Olson [30] for instance identified stress relief as a common game play motive among children, while Mazurek et al. [31] reported similar results in adults with autism spectrum disorder. Reinecke et al. [32] further highlighted the special role of actively playing video games as compared to passive media exposure in eliciting the experience of recovery. This growing interest in playing video games for recreational purposes also becomes apparent in search trends on popular video game distribution services. On the Steam platform for example, users can tag games with specific attributes they believe are appropriate for this game. Frequently applied tags become featured categories in the Steam Store, which can be used by players to discover relevant games. The tag "relaxing" as such a category yields now over 2,800 results in the Steam Store, providing further evidence of relaxation as a common player experience.

## Possible influences of the COVID-19 pandemic on gaming activity

As the COVID-19 pandemic represents a major cause of stress to many people, it is reasonable to believe that there has emerged a growing popularity in playing video games for recreational

purposes during this time. Indeed, reports of changes in leisure activity associated with the pandemic are vast and hint towards a higher interest in gaming. For example, in a sample of 3,052 US adults, Meyer et al. [33] found reduced physical activity and increased screen time. Increased sedentary behaviour was also reported for university students [34]. Naturally, these findings extend to countries all over the world: a decrease of physical activity and increased activities in front of a screen have been shown for adults in India [35], adults and children in China [36, 37], and students in Italy [38].

### The present research

Building upon the above-mentioned empirical findings, the current study seeks to investigate the demand for relaxing video games during the COVID-19 pandemic by analysing player data obtained from the Steam database. Based on reports describing a growing interest in digital activities during the pandemic, it was hypothesized that the peak number of daily players averaged over the time period March to November 2020 increased compared to the average daily player peak (ADPP) from March to November 2019 (i.e., before the outbreak of the pandemic). The daily peak number of players was selected for averaging in order to represent a game's popularity, thereby operationalising demand. As there is evidence that recreation is a desired player experience, particularly in stressful times, a higher rise in ADPPs for relaxation games was expected as compared to games not associated with relaxation. While there is no consent on an exact beginning of the pandemic and countries showing different progressions regarding reported cases and institutional consequences, March 2020 was reported as the beginning of stress-related effects in many studies [2–4] and official public consideration of stress-related concerns made by the World Health Organization (WHO; [39]).

## Materials and methods

### Dataset and preprocessing

The current study hypotheses and the analysis plan were preregistered on OSF. Details on the preregistration can be found under [40]. The dataset used in the current study maps player data from Steam, the largest online game distribution platform for PC games, with over 90 million monthly active users in 2018 and over 20 billion hours of gameplay in 2019 according to Steam's Year in Review reports [41, 42]. The data set was obtained from SteamDB, an independent information service providing data about the applications available on the Steam platform. The information accessible on the website includes for instance statistics about number of followers, number of positive and negative reviews, playtime, and concurrent players for every single day since the release of the game.

The harvested dataset initially contained records from 4,648 of the most played games on Steam. The information included for each game in the initial dataset were name, developer, release date, and the peak number of average daily players (ADPP) for each month since release of the game. For these games, ADPPs were calculated for the time period March to November 2019 (pre-COVID-19 period), and for March to November 2020 (COVID-19 period). Games for which there were missing values in ADPPs for any of the months in the observed time period were eliminated beforehand, resulting in 4,147 remaining applications. In addition, games were deleted from the dataset when released after February 15, 2019, leaving 2,929 games in the dataset. This was done in order to allow for a minimum 2-week warm-up period during which player numbers might not yet be representative of the actual popularity of the game. Furthermore, games with an average peak of less than 3 players per day for the 2019 time period were excluded from the dataset, resulting in 2,379 remaining games.

Games were then selected for the test and control groups. To this end, the existing dataset was augmented with columns containing popular tags and genre for each game. The selection of games followed pre-defined coding criteria and was conducted by both researchers, regularly checking for inter-coder agreement. As an overall criterion, the application had to be a game; demos and other software (e.g., developer tools or utilities) were excluded. For the test group, additional selection criteria were as follows: (1) the game must be tagged as 'relaxing', which included 208 games; (2) a detailed analysis of trailer, game description, and further tags must not lead to the conclusion that the game is not primarily relaxing. To be included in the test group, a game therefore must not show any of the following: (a) Other mood tags, such as 'funny' or 'emotional' are tagged with a higher priority than 'relaxing'. While relaxing games might also include elements that evoke other moods, this was a important step to ensure the experimental group only included games that are prioritise their role in relaxing players over evoking other moods, such as sadness, melancholy, or excitement and humor. After this step, 178 games remained in the test group. (b) Gameplay appears to be action-heavy and intense. After analysing trailer, descriptions, and other promotional material on the Steam Page, games with a high action focus that might be considered relaxing for some, but are generally not designed as a "relaxing game" were excluded. After this step, 169 games remained in the test group. (c) Main elements of the game contain horror, sexual or mature content, or similar characteristics not considered compatible with the main aim of providing a relaxing experience for the player, but rather other emotional aspects that might for some be considered relaxing. Through this procedure, 143 games were identified for the test group (relaxing games). For the control group, 2,124 possible games remained in the dataset. Extreme outliers with more than 7 standard deviations from the mean were further removed from the dataset, leading to 2,216 games in total, with 138 relaxing games, and 2,078 non-relaxing games. It should be noted that test and control group were exclusive, meaning that all games identified as relaxing games were eliminated from the control group pool. This procedure was chosen to have replicable and controlled methods to categorize game characteristics based on how they are presented and described in Steam. Forming groups made it possible to compare a unique subset of games associated with relaxation with a control group to examine effects.

Finally, matching procedures were applied for further statistical analysis. This was done in order to reduce covariate imbalances between test and control group, thereby avoiding a confounded assignment of games to the respective group. Data pre-processed with matching generally produce more robust inferences that are tied to fewer assumptions as compared to non-matched data [43]. The current data set was thus matched based on ADPP in the pre-COVID-19 period (March to November 2019) as the covariate. Matching was achieved using a greedy nearest neighbour algorithm. With this method, each test unit is assigned one control unit, based on the smallest distance in propensity scores between both units. Greedy matching was chosen as it takes the closest match for each value in the test group, which has been shown to perform very efficiently with datasets containing a large pool of control units to chose from [44, 45]. The matching procedure led to a final dataset of 138 games in each the control group and test group. A detailed overview over the selection process can be seen in Fig 1. A preliminary power analysis uncovered the need of 210 participants to reveal a medium-sized effect with a power of $1-\beta = 0.95$. Because non-experimental data needs generally more conservative analyses, the final sample size of 276 was deemed appropriate to reveal a medium-sized effect.

## Analysis

Data analysis was performed with the statistical computing software R. Both parametric and non-parametric analyses of variance were considered for the current dataset. Due to the

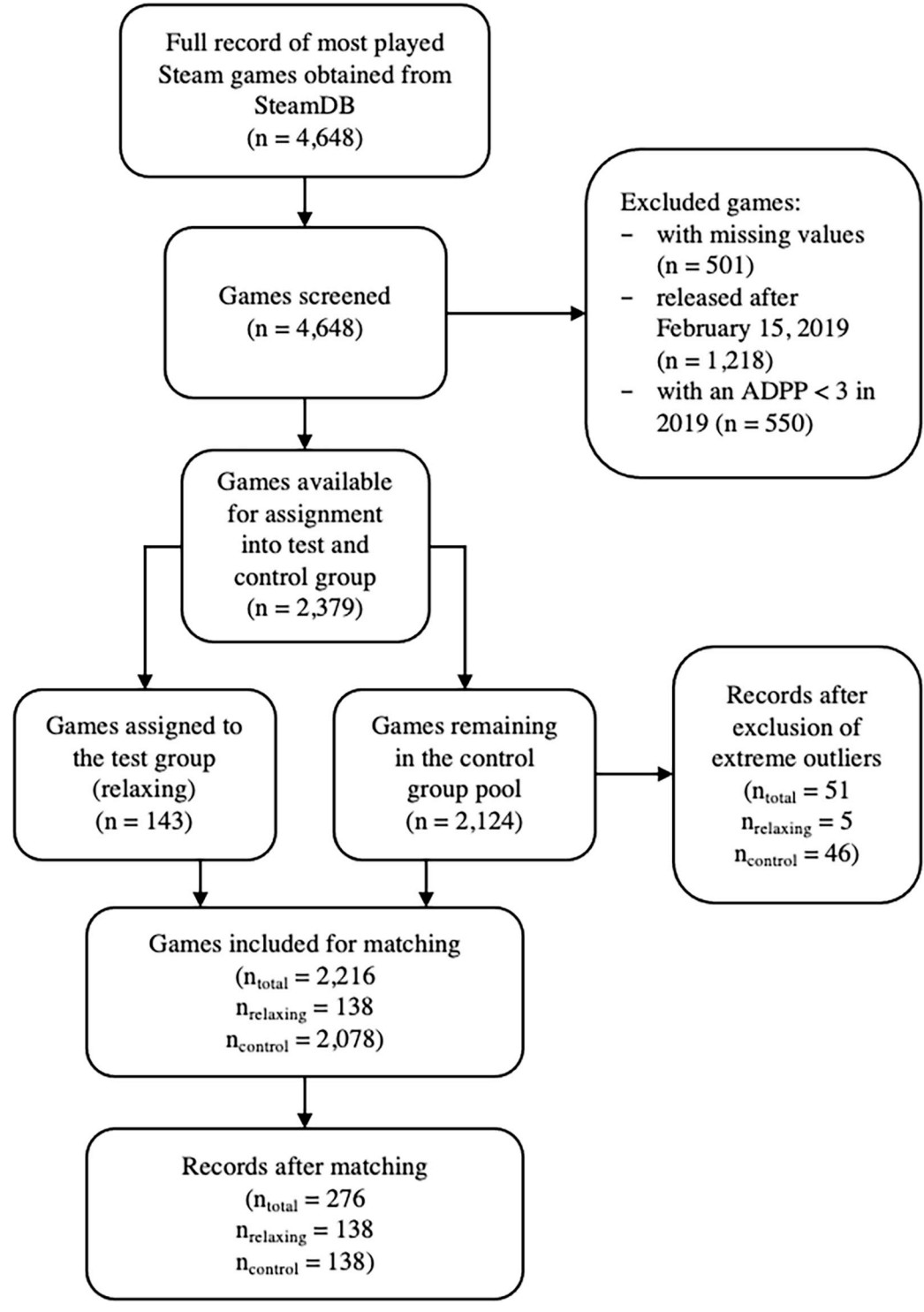

**Fig 1. Flow diagram of the number of games screened and included in the study.**

violation of assumptions of parametric tests (e.g., ANCOVA, or change value ANOVA), non-parametric alternatives such as Quade's test and Wilcoxon signed-rank test were used.

## Ethics statement

Written consent was granted after reviewing the methods of our study by the Physical Sciences Ethics Committee of the University of York in a fast-track procedure. The full statement states: The researchers have taken all reasonable steps to ensure ethical practice in this study and I can identify no significant ethical implications requiring a full ethics application submission to the Physical Sciences Ethics Committee. I have checked and approved all supporting documents required for this application. I understand that completion of this form indicates that from the ethical point of view I am willing to share responsibility for the work being conducted.

## Results

138 games in the relaxing group were matched by the covariate (average daily players for 2019) with 2,077 games in the control group using the MatchIt package [46] in *R*. The final sample consisted of 276 games with 138 games in each group. Descriptive data regarding release years and game genres can be found in Tables 1 and 2. Mean propensity scores for relaxing ($M = 0.06$) and control ($M = 0.06$) groups showed a standardized average mean difference of 0 and a maximal distance of 0.01. Fig 2 shows a visualization of the propensity scores for the matched and unmatched units. Mean covariate value was $M = 226.19$ for the relaxing group, and $M = 226.19$ for the matched control group. A complete overview of descriptive statistics can be viewed in Table 3. Overall, nearest neighbor matching [47] resulted in a well-balanced dataset in regards to covariate similarity (see Fig 3 for the empirical quantile-quantile [eQQ]

**Table 1. Release years of games in the relaxing group and control group.**

| Group | 2008 | 2009 | 2010 | 2011 | 2012 | 2013 | 2014 | 2015 | 2016 | 2017 | 2018 | 2019 |
|---|---|---|---|---|---|---|---|---|---|---|---|---|
| *RelaxingGroup* | 1 | 0 | 2 | 1 | 2 | 9 | 14 | 18 | 19 | 29 | 36 | 7 |
| *ControlGroup* | 1 | 1 | 5 | 8 | 1 | 8 | 15 | 14 | 17 | 36 | 30 | 2 |
| **Total** | **2** | **1** | **7** | **9** | **3** | **17** | **29** | **32** | **36** | **65** | **66** | **9** |

**Table 2. Genres of games for the relaxing group and the matched control group.**

| Genre | Relaxing | Control |
|---|---|---|
| *Action* | 1 | 91 |
| *Adventure* | 31 | 11 |
| *Casual* | 35 | 9 |
| *Horror* | 0 | 5 |
| *Puzzle* | 10 | 0 |
| *Racing* | 1 | 1 |
| *RolePlayingGame* | 5 | 8 |
| *Simulation* | 46 | 3 |
| *Sports* | 2 | 3 |
| *Strategy* | 7 | 7 |
| **Total** | **138** | **138** |

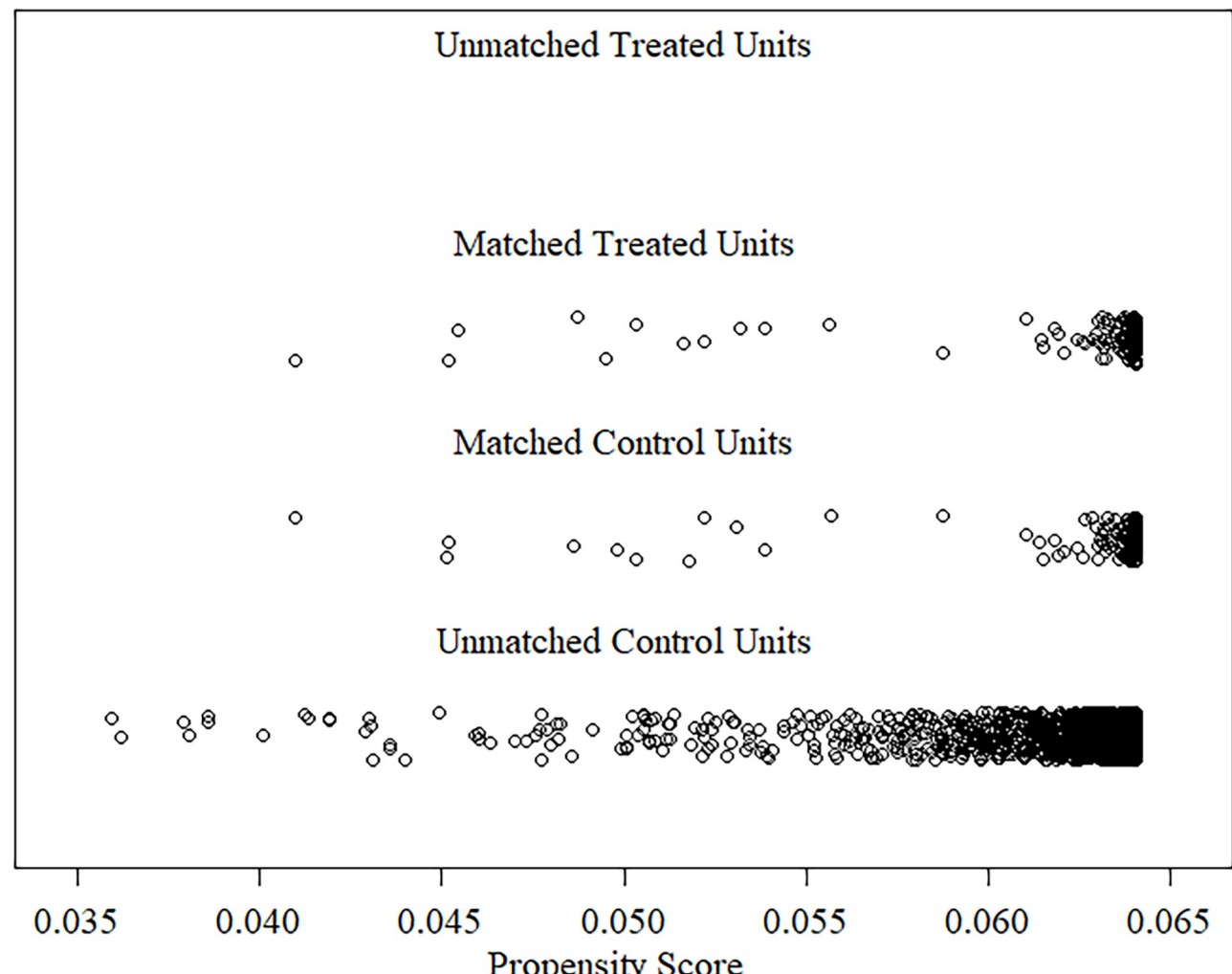

**Fig 2. Propensity scores of matched and unmatched samples.** *R*elaxing games are the treated units and control games are the control units.

plot). However, due to the nature of the data acquisition, the distribution of the covariate was skewed for both groups.

To test whether relaxing games and control games showed a similar expected ADPP progression based on time preceding the pandemic (2015–2019), a preliminary linear regression analysis was conducted, predicting ADPP by time and group. There was no significant main

**Table 3. Descriptive data of ADPPs by group.**

|  |  | 2019 |  | 2020 |  |
| --- | --- | --- | --- | --- | --- |
| *Groups* | *N* | *M* | *SD* | *M* | *SD* |
| *Relaxing* | 138 | 226.19 | 612.16 | 246.54 | 652.76 |
| *Control* | 138 | 226.19 | 657.38 | 240.43 | 657.39 |
| **Total** | **276** | **226.19** | **611.26** | **243.49** | **653.89** |

Sample size (N), mean (M) and standard deviation (SD) for the ADPPs in 2019 and 2020 by groups.

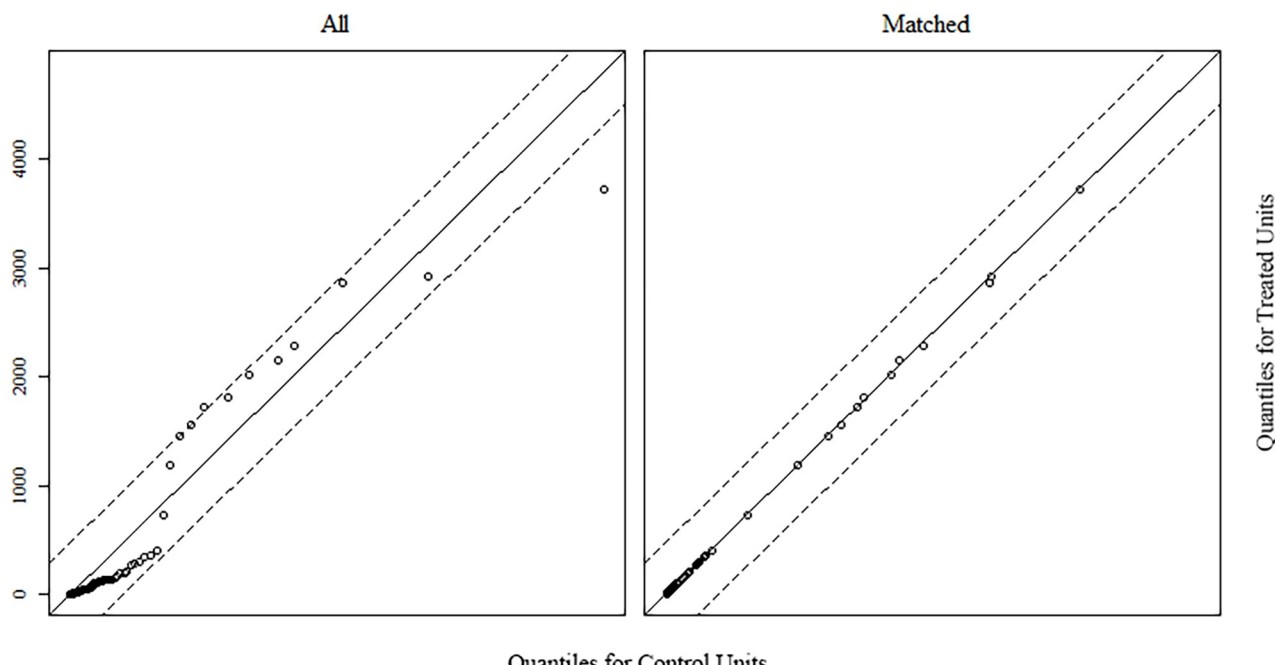

**Fig 3. Empirical quantile-quantile plot of the covariate.** *Depicted are the quantiles of the covariate (ADPPs 2019) for the relaxing and control group before and after matching. Small distances between sample points and the diagonal indicate close similarity.*

effect for time ($t$[752] = 1.67, $p$ = .10) and no interaction effect between time and group ($t$[752] = 0.55, $p$ = .58). A graph of the progression of ADPPs over time by group can be viewed in S1 Fig.

Preliminary tests of ANCOVA assumptions using scatterplots showed a sufficient linear relationship between the independent variable (average daily players for 2020) and the covariate (see Fig 4). Regression slopes for both groups were parallel, pointing towards homogeneity, which was tested using a two-way ANOVA for the interaction between covariate and group ($F$ [1, 272] = 0.05, $p$ = .82).

Results showed no indication for an interaction, so homogeneity of regression slopes was assumed. Just like for the covariate, distributions of the dependents variable and its residuals were highly skewed, which violates not only basic assumptions for ANCOVA, but for a potential change value ANOVA, which is commonly used as an alternative to present average treatment effects on treated (ATTs), i.e. in this case the gain of average daily players of the relaxing games group specifically between 2019 and 2020.

It was therefore decided to explore the overall time effect and the ATT using more robust non-parametric methods that are not distribution dependent. First, the overall time effect (the increase in ADPPs for both groups combined between 2019 and 2020) was calculated using a Wilcoxon signed-rank test [48]. The test showed a significant difference between 2019 and 2020 ($Z$ = 4.70, $p$ <.01). So, between March 2020 and October 2020, a significantly higher daily average peak per game was observed than for the same period in 2019. Mean ADPPs increased from $M$ = 226.19 in 2019 to $M$ = 243.49 in 2020 (see Table 3 for more information). An effect size was calculated to quantify the interpretation of the effect after Rosenthal [49]. Using Cohen's criteria [50], the calculated effect size ($r$ = 0.20) can be considered a medium-sized effect.

To test if this increase was even larger for the relaxing games compared to the control games, the ATT was calculated using Quades's rank analysis of covariance [51]. This method

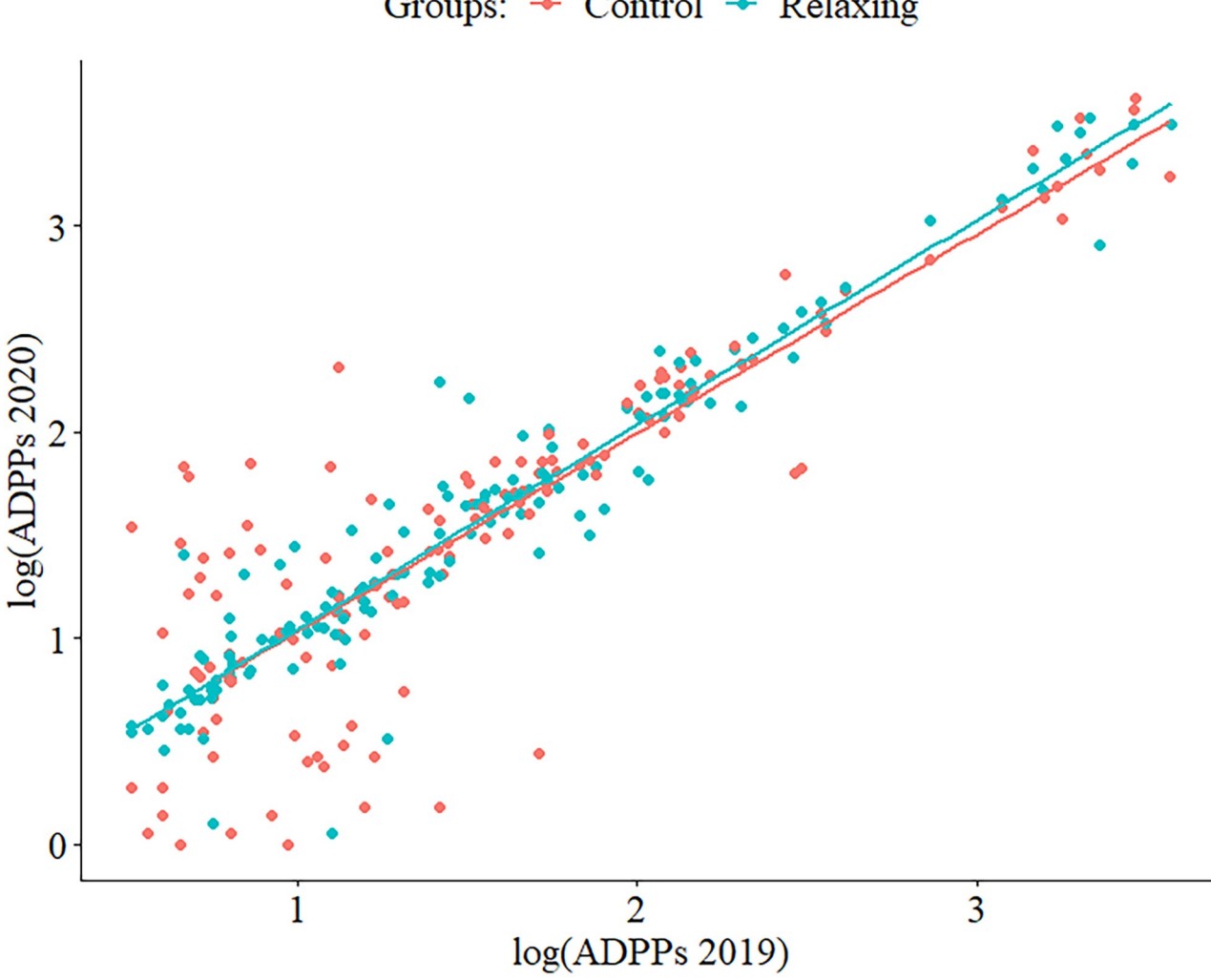

**Fig 4. Scatterplot of the relation between ADPPs in 2019 and ADPPs in 2020 by groups.** Scales have been log-transformed for clearer data presentation.

utilises the residuals of the regression of ranked dependent variables and ranked covariates, rather than the non-normal distributed variables themselves. Results showed no significant group effect ($F[1, 67] = 0.05$, $p = .88$), meaning the ATT was not significantly different from the average treatment effects on the control group. In other words: No significant difference of average daily player increase was observed between the relaxing game group and the control group. A boxplot visualising the group differences can be seen in Fig 5.

## Discussion

To examine the increase of demand for video games associated with relaxation during the COVID-19 pandemic, relaxing games and matched non-relaxing games have been compared in regards to their average daily player peaks for the periods of March to October in 2019 and 2020 respectively. Results revealed a medium-sized increase of ADPPs over both groups for the COVID-19 period compared to the pre-COVID-19 period, but no significant differences between both groups regarding this increase.

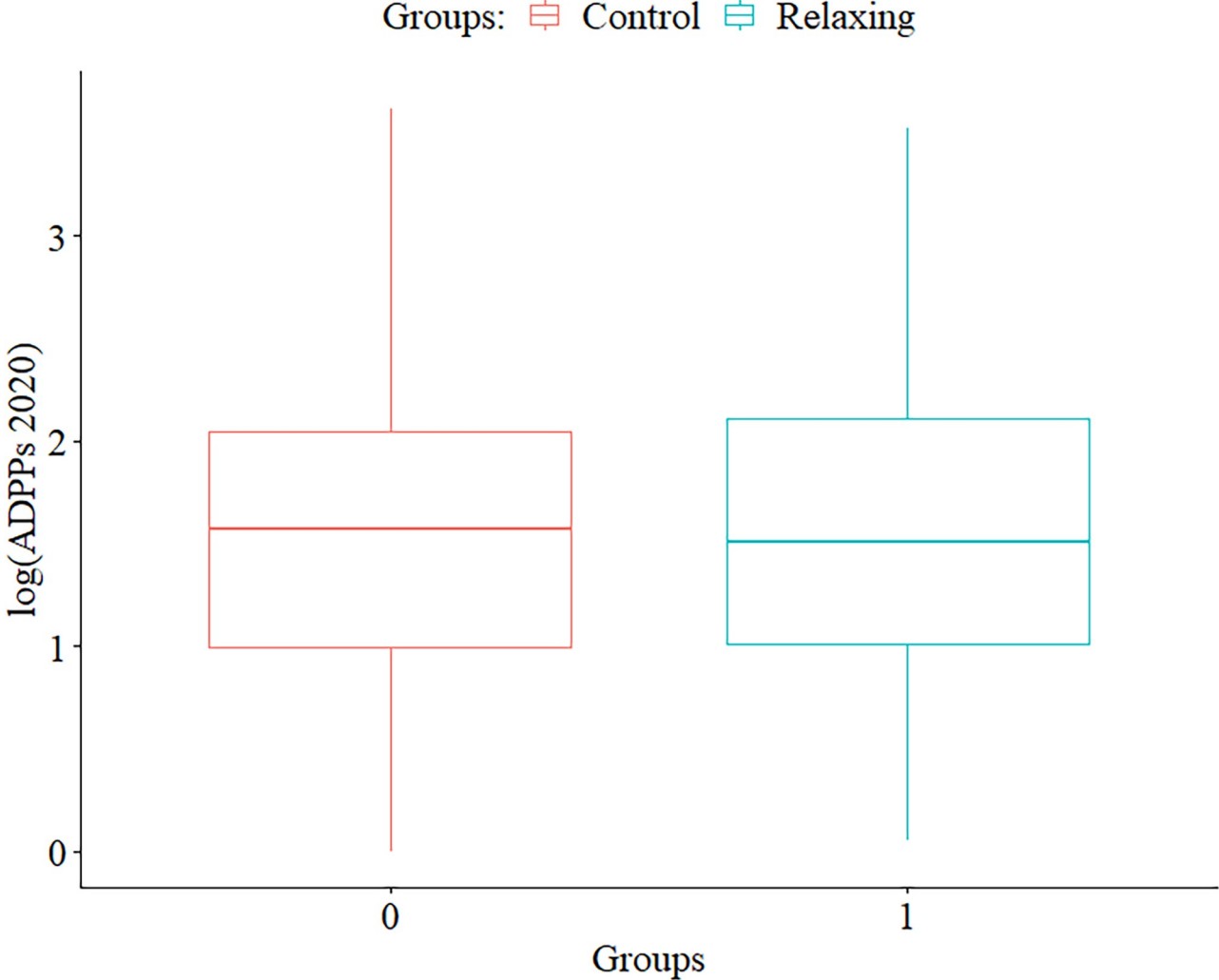

**Fig 5. Boxplots of the differences in ADPPs in 2020 by groups.** Scales have been log-transformed for clearer data presentation.

While the hypothesized overall increase in ADPPs was found, there was no evidence to support the hypothesized special role of relaxation related games during the pandemic. There are various implications for these findings that possibly uncover common misunderstandings regarding relaxation-seeking in very stressful times and individual reasons to play video games of certain styles.

### The role of relaxing games in increased daily player peaks

The COVID-19 pandemic has vast influences on people's everyday behaviour and the presented increase of ADPPs in the current study mirrors the current findings in the literature, including the findings of the increase of screen time all over the world (e. g. [33–36, 38]). During the months of lockdown, availability for outdoor activities was very limited and digital alternatives, including video games, have experienced a rise in popularity. In fact, there is no

reason to doubt an overall increase in digital activities, but there is disagreement regarding the reasons and consequences of these behaviours, specifically for video games.

Just as there is not much doubt about the behavioural changes that resulted from the pandemic, the literature provides a huge amount of convincing evidence regarding the changes relating to mental health, specifically psychological stress [2–4]. Multiple studies associated the COVID-19 related increased screen time and decreased physical activity with mental health problems (e.g. [52, 53]). The pandemic itself and consequences of the lockdowns for people's lifestyles have been identified as risk factors for psychological problems. However, more and more recent studies provide arguments that link the role of video games in specific not to a mental health danger, but to mental health improvement and protections against such risk factors (e.g. [19, 20, 25]). It further seems that people are more and more drawn to video games as a way to actively recuperate from stressful events [28, 29]. While the present findings do not hold any information about why people play more during the pandemic, the increased numbers of daily players across all games do not contradict this particular view. It is however important to note that this overall increase might not be related to COVID-19 but rather represent a naturally expected increase of ADPPs independent from the pandemic, but possibly related to factors such as reduced cost or increased accessibility of technology. As S1 Fig shows, ADPP peaks for most game seem to lie in 2018 with a big decrease in ADPPs for all games between 2018 and 2019. It is very much possible that the pandemic had an effect on this trajectory from 2019 onwards that seems to invert a decreasing trend, its significance does however appear insignificant considering the the overall variation in ADPPs over time and the unexplained decrease of ADPPs after 2018, which might represent multiple reasons, such as a natural decrease in interest after a few years post-release. In any case, the data does not allow for conclusive attribution of observed trends. Not only is it questionable to assume the pandemic had a large effect on player increase, the presented data shows no support for the assumption that that people would play even more games associated with relaxation to cope with the stress caused by the pandemic.

Therefore, the overall effect might just be a product of natural interest variation, although more available free time during the pandemic might also explain some of the variance reversing the declining trend on ADPPs over time. Due to lockdown restrictions more time is spent at home, so it is quite natural to expect more time spent on common indoor activities, such as video games. While it makes sense to think of video games as ways to counter the mental strain caused by the pandemic, it is important to control for variables like increased boredom due to the restrictions. It might even be argued that natural interest decline can be seen as a simpler explanation, especially considering that no difference between relaxing games and control games was found. Future research investigating the overall rise in video game player peaks should therefore make sure to control for variables like boredom and natural interest decline to better explore how much video games are actually played for the potential mental health benefits. It might be that the vast majority of players do not play video games as a way to relieve stress, but rather because there is not much else to do. If this is the case, an important follow-up question would be: Why are people not playing more as a way to decrease stress? This is a question that has many potential implications for the game industry and players alike. As the view of video games as a way to reduce stress is quite a new direction, it could be that people are just not aware of the potential positive effects or that these effects are not convincing enough to change actual behaviour. Another explanation would be the perception of the current game catalogue that is considered "relaxing". There might be limited appeal of this type of game that includes mostly simulation, adventure, and casual games (see Table 2), even if these games are seen as emotionally profitable. In summary, there is definitely room to research these assumptions to get a more complete picture of how games are viewed and why games are

played, which also might hold further information about potential strategies in marketing and designing games specifically for relaxation (i.e. serious games). It is however important to note that the present findings do not hold strong evidence for or against any theory on why people are playing games, so further research is clearly necessary to explore this.

## Means of relaxation

It might therefore still be true that stress relief is an important selling point for video games, especially in very stressful times. If this is the case, another perspective to explain the data might emerge if relaxation from video games is not seen as specifically seeking out relaxing video game elements and styles, but relaxing from playing any video game, regardless of style or genre. A wide variety of individual differences regarding physiological stress-responses have been reported [54] and emotion regulation technique effectiveness is also known to be highly dependent on individual differences [55, 56]. Game elements that consistently fall under the definition of "relaxing" might therefore not mirror what individual people find relaxing and seek out. One possible counter-argument could be the amount of mostly violent action games in the control group (see Table 2), which may not help to reduce stress, but rather cause it (e.g. [57, 58]). Recently, more research has been conducted that demonstrates the importance of individual differences in response to violent games (e.g. [59, 60]) and even shed light on important elements with a positive relation to mental health. For example, Collins and Cox [61] found that action games and shooters have the largest effect on after work recovery than any other genre. Moreover, social interactions and support have been identified as important factors moderating positive effects of video games on mental health [60, 61]. In fact, social interactions have consistently been shown to be an important buffer against the negative impacts of stress (e.g. [62]), which also relates to social components within otherwise violent video games. Maroney et al. [63] for example argue that all types of games provide means to reduce negative states, depending on individual stress-coping mechanisms, and provide again more evidence for the importance of social interactions as a mediator for these effects. As COVID-19 does not only lead to more stress, it also leads to less social interactions, so it makes sense that there is an increased demand for games with social components, which also might lead to a more regulated stress-response. In other words, we might not just relax with games that are specifically made for relaxation purposes.

This might be a viewpoint that should be considered when reviewing the role of playing video games during the COVID-19 pandemic. In current research, a high emphasis is put on specific games with relaxing elements, and how these elements could be associated with well-being. Specifically games like Animal Crossing: New Horizons have been used as an example of the possible effects of video games during the pandemic [64, 65]. While this study does not invalidate this approach, it might be worthwhile to expand the research to include all kinds of games, game elements, styles, and genres to better understand how video games as a whole can be used to fight stress and increase mental health, especially during a highly straining period like the COVID-19 pandemic.

## Limitations and future directions

While this study does provide many indications of the role of video games during the COVID-19 pandemic, it is important to note that it does not allow for definite conclusions. It is possible that none of the provided explanations are sufficient to understand the precise nature of why there was no difference between the groups. However, it can be argued that the possibilities provided are relevant research topics that are worthwhile to investigate. In fact, one of the biggest findings is the lack of clarity regarding the results. On the one hand, it is still perfectly

reasonable to believe that video games have an important role in relieving COVID-19-related stress, while on the other hand there seems to be a gap between theoretical assumptions and actual player behaviour that needs to be filled by experimental studies. So, while no major conclusions can be drawn, this study can be seen as an indication of the importance of further research that includes actual player behaviour.

Additionally, some limitations have to be considered that emerged as a consequence of the study design. As no randomization was possible and a quasi-experimental design was chosen, it might be the case that a naturally lower statistical power could have influenced the reported results. Measures like the reported effect size of the overall ADPP increase must be interpreted with caution, as these could behave differently (and often more conservative) than the respective measures for parametric tests. The limited sample size also limits the probability of revealing very small effect sizes that still could be relevant for data containing video game players all over the world. To have a more precise picture of the discussed effect, a large-scale experimental study would be appropriate, although the informational gain might not be worth the effort. Another characteristic of this study is the restriction to Steam and to a certain player base. Steam is only a part of the whole video game industry and does not reflect players on consoles or mobile devices. Because relaxing games were limited in ADPPs and controls were matched, most of the most popular games with hundreds of thousands of daily players had to be excluded. As a result, this study provides insights about a specific type of game and may not be generalizable for other platforms. It also does not reflect types of players, which could also be an interesting target for further examination. Rather, this study explores the effect of game characteristics and has limited implications for overall player behaviour, as observations were games and not individual players. Methodological limitations are however a consequence of this data-driven approach, which is essential to make sense of the theoretical assumptions that are more and more emerging. These issues can only be solved by conducting even more empirical studies, in the best case experimental studies, to really inform about how well certain theories translate to the real world. Because the games were categorized into fixed groups with considerable in-group variance, it becomes also apparent that game research is in need of more work towards game aspect classification. For example, relaxation could be seen as a continuous dimension, representing players' perceived value of having a relaxing experience with a game, which would open the door to new analysis tools, such as linear regression models. Before this is possible, obtainable and valid measures need to be established to turn game characteristics into valid constructs for experimental data analysis. The present study provides an example of how to form controlled and distinct groups that can be compared, which gives more insight into differences and similarities, but is restricted to the nature of the groups.

Overall, it is important to view this study as part of the evidence exploring the current link between theoretical frameworks and real-life data in the research about the mental health benefits of video games. There are limits with respect to generalization and it would be misguided to conclude that the demand of relaxing games is currently equal to the demand of non-relaxing games. Instead, the results should be interpreted in regards to their informational value when it comes to the gaps that are currently present within this field. More data-driven approaches should be applied to address these gaps and connect the various theories with actual gaming practices.

## Supporting information

**S1 Fig. Mean ADPP by time for relaxing games vs control games.**
(TIF)

## Acknowledgments

We thank Dr. David Zeindle (University of York) for support and also the continuous efforts of the team behind SteamDB for data acquisition.

## Author Contributions

**Conceptualization:** Maximilian Croissant.

**Data curation:** Maximilian Croissant.

**Formal analysis:** Maximilian Croissant.

**Investigation:** Madeleine Frister.

**Methodology:** Maximilian Croissant.

**Writing – original draft:** Maximilian Croissant, Madeleine Frister.

**Writing – review & editing:** Maximilian Croissant, Madeleine Frister.

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
