## [Decision Letter · Decision Letter 0]

11 Oct 2021

PONE-D-21-07746A data-driven approach for examining the demand for relaxation games on Steam during the COVID-19 pandemicPLOS ONE

Dear Dr. Croissant,

Thank you for submitting your manuscript to PLOS ONE. After careful consideration, we feel that it has merit but does not fully meet PLOS ONE’s publication criteria as it currently stands. Therefore, we invite you to submit a revised version of the manuscript that addresses the points raised during the review process.

The points that need more attention are related to methodological controls. Please submit your revised manuscript by Nov 25 2021 11:59PM. If you will need more time than this to complete your revisions, please reply to this message or contact the journal office at plosone@plos.org. Please include the following items when submitting your revised manuscript:A rebuttal letter that responds to each point raised by the academic editor and reviewer(s). You should upload this letter as a separate file labeled 'Response to Reviewers'.A marked-up copy of your manuscript that highlights changes made to the original version. You should upload this as a separate file labeled 'Revised Manuscript with Track Changes'.An unmarked version of your revised paper without tracked changes. You should upload this as a separate file labeled 'Manuscript'.

We look forward to receiving your revised manuscript.

Kind regards,

Flávia L. Osório, PhD

Academic Editor

PLOS ONE

Additional Editor Comments (if provided):

the points that need more attention are related to methodological controls

Journal Requirements:

2. Please note that in order to use the direct billing option the corresponding author must be affiliated with the chosen institute. Please either amend your manuscript to change the affiliation or corresponding author, or email us at plosone@plos.org with a request to remove this option.

Reviewers' comments:

Reviewer's Responses to Questions

**Comments to the Author**

1. Is the manuscript technically sound, and do the data support the conclusions?

Reviewer #1: Partly

Reviewer #2: Yes

2. Has the statistical analysis been performed appropriately and rigorously? 

Reviewer #1: Yes

Reviewer #2: I Don't Know

3. Have the authors made all data underlying the findings in their manuscript fully available?

Reviewer #1: Yes

Reviewer #2: Yes

4. Is the manuscript presented in an intelligible fashion and written in standard English?

Reviewer #1: Yes

Reviewer #2: Yes

5. Review Comments to the Author

Reviewer #1: Review of “A data-driven approach for examining the demand for relaxation games on Steam during the COVID-19 pandemic”

Manuscript Overview: The authors strive to test two hypotheses: I) that gaming became more prevalent during the pandemic (as measured by average daily player peak) and II) that this was particularly true for “relaxing” video games. Using data from Steam (the largest online video game distribution platform) the authors created a matched set of 138 relaxing games and 138 non-relaxing games. Mean average daily player peak increased from 2019 to 2020 for both types of games (supporting the first hypothesis), but did not do so in a differential manner (failing to support the second hypothesis).

Comments/questions:

1) My biggest issue/question is that the authors’ questions are all with regard to a change in time (i.e., changes from 2019 to 2020) with the specific thought that the pandemic (2020) might be been “different.” However, they don’t have an appropriate baseline to examine what type of change would have been expected in the absence of a global pandemic.

As an example, the percentage of self-identified Democrats who believe that racial discrimination in the United States significantly hinders equality rose from 64% in 2019 to 68% in 2020. While it’s theoretically possible that the increase was related to something that happened in 2020, that possibility becomes much less likely when you look at the changes from the previous years (i.e., 62% in 2018, 57% in 2017, 52% in 2016, etc.). When the historical data is available, it’s clear that 2020 fits pretty nicely in a long-term monotonic trend. There’s no disjunction as would be predicted if something that occurred at the start of 2020 (noting that all of the data above was collected prior to the murder of George Floyd, which may well create a disjunction in 2021).

This is obviously important in interpreting the current data. Was the increase in ADPP “unexpected” given long-term trends? What about the increase per game type? For instance, it’s certainly possible that the “expected” long-term trend for relaxation games would be to decrease from year to year, while the expectation for the control games would be an increase. If that were true, then the current findings (where both increased by a similar amount), would have a *very* different interpretation.

So, in all, in order for the results to be interpretable/meaningful, the authors really need to move toward an analysis strategy where they examine potential deviations from an expected trajectory from 2019 to 2020 (and obviously by going back through time, it would be possible to model whether any deviations, if found, are “unusual” deviations).

2) Asking the particular question the authors are interested in is obviously super strongly predicated on the ability to appropriately categorize games.

As such, I think:

a. The relevant section in the methods would benefit from additional detail.

i. For instance, it would be typical in coding this sort of dataset to indicate how many games failed each of the various criteria (i.e., how many games had a “relaxing” tag, but then visual inspection suggested that they were “action-heavy”).

ii. Similarly, explanation for the criteria would also be useful (i.e., it’s unclear to me why it’s a problem if a relaxing game is also funny?).

b. It would also be worth walking through why the appropriate contrast is between relaxing games and “non-relaxing” games. It seems like the stronger test of (what at least I view as the core hypothesis) would pit relaxing games against stressful games (i.e., testing relaxing versus everything else is a weaker test than pitting relaxing versus the opposite of relaxing on the dimension of interest).

3) The authors will want to justify starting with March 2020 since (globally speaking) that is a bit “in-between” with respect to the start of the pandemic (e.g., where I am in the United States, nothing really changed at all until the third week of March 2020 – and so using March in the data set is weakening the data set).

Reviewer #2: Thank you so much for this research. Your paper adds, as you say, real-life data to explore theories on the use/impact of serious games. I can't speak to the statistical analysis part.

What went well: Your discussion section answered all the critics I had. For instance; you didn't assume that relaxation games were the solution. People play video games to relax regardless of style or genre; and stress- reducing activities are different for everyone.

Questions I still have:

Your findings point out that more people are playing more games and this increased prior to COVID. Now does this coincide with anything else happening in the game environment? Cost? Improved technology? Improved internet access?

You ask: Why are people not playing more as a way to decrease stress? Consider that the broader messages are telling us to get up and move, go out for a walk etc. Gaming is part of our 'feel good' activities, but one needs a break from staring at a screen regardless of how stress-reducing it is.

Your paper is well written, well argued. My two questions fit into how does gaming fit into the broader context. They stem from curiosity more than criticism so don't feel pressured to change anything in the paper.

6. PLOS authors have the option to publish the peer review history of their article (what does this mean?). If published, this will include your full peer review and any attached files.

Reviewer #1: No

Reviewer #2: No

---

## [Author Response · Author response to Decision Letter 0]

1 Nov 2021

Thank you so much for your consideration of our study “A data-driven approach for examining the demand for relaxation games on Steam during the COVID-19 pandemic” and the opportunity to revise the manuscript based on constructive feedback.

We are very thankful for both reviewers and their thoughtful comments. We’re very happy that both reviewers agree that this study potentially provides a worthwhile contribution and found that the presentation, data availability, and statistical analyses all meet the necessary requirements for publication. Even more happy are we for the very well argued and illustrated suggestions that were presented to improve the manuscript. We have included a detailed response to every comment that goes into more detail how we approached the changes and the nature of the changes that we have made and that can be viewed in the new manuscript, including track changes. We especially hope that we addressed the main issue of reviewer 1 regarding the methodological approach of the study. We included additional analyses to provide further justification of our approach that include the proposed historical data to show that our two groups can be meaningfully compared to draw conclusions.

We also apologize for not sufficiently having addressed the additional journal requirements in the original submission. We hope that the revision reflects all style requirements. We also included the full ethics statements in the methods section of the manuscript. If there are still any concerns or questions, please do not hesitate to approach us. Thank you very much.

Response to the reviewer comments on manuscript PONE-D-21-07746 

“A data-driven approach for examining the demand for relaxation games on Steam during the COVID-19 pandemic”

Reviewer #1

The authors strive to test two hypotheses: I) that gaming became more prevalent during the pandemic (as measured by average daily player peak) and II) that this was particularly true for “relaxing” video games. Using data from Steam (the largest online video game distribution platform) the authors created a matched set of 138 relaxing games and 138 non-relaxing games. Mean average daily player peak increased from 2019 to 2020 for both types of games (supporting the first hypothesis), but did not do so in a differential manner (failing to support the second hypothesis).

First of all, we are very thankful for this thoughtful and insightful review that offers some perspectives we missed. We hope to address all concerns in the revised manuscript.

1) My biggest issue/question is that the authors’ questions are all with regard to a change in time (i.e., changes from 2019 to 2020) with the specific thought that the pandemic (2020) might be been “different.” However, they don’t have an appropriate baseline to examine what type of change would have been expected in the absence of a global pandemic.

As an example, the percentage of self-identified Democrats who believe that racial discrimination in the United States significantly hinders equality rose from 64% in 2019 to 68% in 2020. While it’s theoretically possible that the increase was related to something that happened in 2020, that possibility becomes much less likely when you look at the changes from the previous years (i.e., 62% in 2018, 57% in 2017, 52% in 2016, etc.). When the historical data is available, it’s clear that 2020 fits pretty nicely in a long-term monotonic trend. There’s no disjunction as would be predicted if something that occurred at the start of 2020 (noting that all of the data above was collected prior to the murder of George Floyd, which may well create a disjunction in 2021).

This is obviously important in interpreting the current data. Was the increase in ADPP “unexpected” given long-term trends? What about the increase per game type? For instance, it’s certainly possible that the “expected” long-term trend for relaxation games would be to decrease from year to year, while the expectation for the control games would be an increase. If that were true, then the current findings (where both increased by a similar amount), would have a *very* different interpretation.

So, in all, in order for the results to be interpretable/meaningful, the authors really need to move toward an analysis strategy where they examine potential deviations from an expected trajectory from 2019 to 2020 (and obviously by going back through time, it would be possible to model whether any deviations, if found, are “unusual” deviations).

Thank you for this comment. This is an important point that we don’t want people to be confused about. Our main response here lies in how we define our research question. We now realize that it was a bit misrepresented in the original manuscript. Our main research question is not concerned with testing how much influence the pandemic had on player increase. You are right in saying that this cannot be sufficiently answered without examining what increase we would normally expect over time. Looking at a long-term trend would be a worthwhile contribution, but this was not the main hypothesis we wanted to explore. We are concerned with how ADPPs development differ between an experimental group (relaxing games) vs. a control group, i.e., is there a special increase in attention for relaxing games vs. other games during the pandemic?

If all games get naturally more playtime between 2019 and 2020, our main question if relaxing games are (for whatever reason) special in this increase can still be answered, even if we assume the pandemic had no influence on overall player base increase. In this context, it becomes less important to know how the pandemic influences the number of players: Whatever the influence is, it is not uniquely different for relaxing games. 

This is also why we chose this specific control group and no other group that relates to how relaxing or not-relaxing a game is. This control provides us with the expectation we need to statistically test differences. For your example, this would be if a specific subgroup of Democrats had a different increase in this beliefve between 2019 and 2020 compared to any other Democrat - regardless of the reasons or time trajectory. Theories provide explanations of why relaxing games are more played during the pandemic, but we found in fact no evidence for that. You are right in saying that we can’t even say if games in general had a pandemic-related player increase, which in our opinion does strengthen the notion further that relaxing games do not play a special role in times of the pandemic. 

This explanation of course only holds true if we can assume that relaxing games have a similar “trajectory” as all other games. Our assumption was that this would be a natural consequence of our methodology: All games are part of the same population (specifically part of games available on the Steam online store) and to have a baseline for expected trajectory, we tested our hypothesis 1: Over all games (which is relaxing and all other games), there is an increase, so this increase was our baseline. Our strategy was to compare relaxing games to a control, that are both created from the same population - a population that provided our baseline in hypothesis 1, so we knew what to expect for ADPP change in 2019 vs. 2020 for games in general. 

We now realize that this might not hold true in every case, so we added historical data for the games to our analyses to test if relaxing games and control games ADPPs developed similarly over time. We added the following paragraph in our Results section:

“To test whether relaxing games and control games showed a similar expected ADPP progression based on time preceding the pandemic (2015-2019), a preliminary linear regression analysis was conducted, predicting ADPP by time and group. There was no significant main effect for time (t[752] = 1.67,p= .10) and no interaction effect between time and group (t[752] = 0.55,p= .58). A graph of the progression of ADPPs over time by group can be viewed in S1 Fig.” (lLines 192-197) 

We hope that this would provide enough evidence to assume that relaxing games and control games stem indeed from the same population, which is necessary for our other analyses. We matched control games based on their 2019 performance - so all games start at the same point. The difference between trajectories for relaxing games and control games can therefore be reflected in our ANCOVA, so we can make conclusions about if this subgroup behaves differently than our expected outcome. 

This explanation is very relevant for our second (and main) hypothesis. Our first hypothesis was tested not to show that COVID increased ADPPs over all games, but to create a baseline - to create the expected trajectory. It is very true that our first hypothesis cannot be meaningfully related to the pandemic. This would be an interesting question to ask, but we fear that it would distract from our main point and further complicate analyses and therefore possible interpretations. We hope that the main message taken from our study would show the gap between theoretical work and supporting data. 

To address all your concerns appropriately, we’ve made further changes to the manuscript. We included an illustration of the expected historic trajectory (2015-2020) for our dataset and added this in the supplementary material (see S1 Fig). Most importantly, we found no difference in group trajectory. We also did not find an overarching time effect, this has to be however taken cautiously as our regression does simplify all possible contributions to ADPP change. Our illustration shows that there could be a complex association between time and other related factors (such as technology development) that might be intriguing to research further, but we did not want to shift the focus and overcomplicate our findings. Rather, we used the data and the figure to validate our assumption about expected trajectory and further explain how our study fits into the research of COVID-19 related effects on player base change, so we added the following section to the discussion:

“While the present findings do not hold any information about why people play more during the pandemic, the increased numbers of daily players across all games do not contradict this particular view. It is however important to note that this overall increase might not be related to COVID-19 but rather represent a naturally expected increase of ADPPs independent from the pandemic, but possibly related to factors such as reduced cost or increased accessibility of technology. As Fig S1 shows, ADPP peaks for most games seem to lie in 2018 with a big decrease in ADPP for all games between 2018 and 2019. It is very much possible that the pandemic had an effect on this trajectory from 2019 onwards that seems to invert a decreasing trend, its significance does appear however insignificant considering the the overall variation in ADPPs over time and the unexplained decrease of ADPPs after 2018, which might represent multiple reasons, such as a natural decrease in interest after a few years post-release. In any case, the data does not allow for conclusive attribution of observed trends. Not only is it questionable to assume the pandemic had a large effect on player increase, the presented data shows no support for the assumption that that people would play even more games associated with relaxation to cope with the stress caused by the pandemic.“ (lLines 264-280)

We also changed some of the reasoning in the literature and discussion sections, further eliminating confusion about how the main effect could point to pandemic-related influences vs. random observation within a time frame, as the data does not allow for such inferences. We hope these changes are sufficient to underline the main message our study provides.

2) Asking the particular question the authors are interested in is obviously super strongly predicated on the ability to appropriately categorize games.

As such, I think:

a. The relevant section in the methods would benefit from additional detail.

i. For instance, it would be typical in coding this sort of dataset to indicate how many games failed each of the various criteria (i.e., how many games had a “relaxing” tag, but then visual inspection suggested that they were “action-heavy”).

Thank you for this suggestion. We tried to make our selection process as easily understandable and replicable as possible. In line with your suggestion, we included the exact number of games that were excluded through every step of our selection process. This includes the following sections:

“(1) the game must be tagged as ‘relaxing’, which included 208 games;” (line 121)

“(a) Other mood tags, such as ‘funny’ or ‘emotional’ are tagged with a higher priority than ‘relaxing’ […]. After this step, 178 games remained in the test group.” (line 124)

“(b) Gameplay appears to be action-heavy and intense. […]. After this step, 169 games remained in the test group.” (line 129)

“(c) Main elements of the game contain horror, sexual or mature content, or similar characteristics […]. Through this procedure, 143 games were identified for the test group.” (line 133) 

For more details see lines 121-138 in the manuscript. 

ii. Similarly, explanation for the criteria would also be useful (i.e., it’s unclear to me why it’s a problem if a relaxing game is also funny?).

This was also reworked in the manuscript. We added further information about our inclusion criteria and detailed reasoning. The main aim of the study was to compare relaxing games as a type of game and not just a game that is by a certain number of people considered a relaxing experience. Because of that, it was important to us to separate it from other target experiences and moods, if these other moods had a higher priority over “relaxation”. Further explanation of these criteria includes the following additions:

“(a) Other mood tags, such as ‘funny’ or ‘emotional’ are tagged with a higher priority than ‘relaxing’. While relaxing games might also include elements that evoke other moods, this was an important step to ensure the experimental group only included games that are prioritize their role in relaxing players over evoking other moods, such as sadness, melancholy, or excitement and humor.” (lines 124-129)

“(b) Gameplay appears to be action-heavy and intense. After analyzing trailer, descriptions, and other promotional material on the Steam Page, games with a high action focus that might be considered relaxing for some, but are generally not designed as a "relaxing game" were excluded.” (lines 129-134)

“(c) Main elements of the game contain horror, sexual or mature content, or similar characteristics not considered compatible with the main aim of providing a relaxing experience for the player, but rather other emotional aspects that might for some be considered relaxing.” (lines 133-137)

b. It would also be worth walking through why the appropriate contrast is between relaxing games and “non-relaxing” games. It seems like the stronger test of (what at least I view as the core hypothesis) would pit relaxing games against stressful games (i.e., testing relaxing versus everything else is a weaker test than pitting relaxing versus the opposite of relaxing on the dimension of interest).

This is a very valid point that was part of many discussions of the initial design of the experiment. There are however multiple problems that would limit the validity further in our opinion. While the experience of relaxation could arguably be seen as a dimension of continuous data, there is no real way of mapping all Steam available games onto this dimension. Rather, tags and available game aspects were used to categorize games into “genre”-style categories to form groups. This provided the ability to statistically compare the group of relaxing games with a sound control group of other games. As we described in our response to your first concern, this control group represents our expected outcome for all games, as shown for the first hypothesis. We agree that the comparison of relaxing vs. specifically stressful games would be an interesting addition, but creating a “stressful” category proved very difficult with our approach and would need more elaborate coding criteria. This in turn would open many more questions regarding sufficient control groups for both newly formed groups and in our opinion would have ultimately distracted from the main take-away, which is: Are games that can be identified as being part of the “relaxing games” category any different than other games? We see this question as an important and non-trivial contribution to many questions, including how game characteristics research could be conducted and valid (while also achievable with accessible data). To do this, group formation was necessary. Having multiple control groups would have strengthened the approach, but there are no Steam tags that represent the exact opposite of “relaxation”, so we opted for a traditional control condition. Forming more groups or finding ways to operationalize characteristics on a continuous dimension would be a next logical step. To better address possible concerns and justify our approach, we added two new paragraphs to the document:

“This procedure was chosen to have replicable and controlled methods to categorize game characteristics based on how they are presented and described in Steam. Forming groups made it possible to compare a unique subset of games associated with relaxation with a control group to examine effects.” (lLines 143-146)

“These issues can only be solved by conducting even more empirical studies, in the best case experimental studies, to really inform about how well certain theories translate to the real world. Because the games were categorized into fixed groups considerable in-group variance, it becomes also apparent that game research is in need of more work towards game aspect classification. For example, relaxation could be seen as a continuous dimension, representing players' perceived value of having a relaxing experience with a game, which would open the door to new analysis tools, such as linear regression models. Before this is possible, obtainable and valid measures need to be established to turn game characteristics into valid constructs for experimental data analysis. The present study provides an example of how to form controlled and distinct groups that can be compared, which gives more insight into differences and similarities, but is restricted to the nature of the groups.” (lLines 382-393)

3) The authors will want to justify starting with March 2020 since (globally speaking) that is a bit “in-between” with respect to the start of the pandemic (e.g., where I am in the United States, nothing really changed at all until the third week of March 2020 – and so using March in the data set is weakening the data set).

Thank you for this comment. This is an important point that needs to be better highlighted. The COVID pandemic was (and still is) not a continuous event that had not the same consequences for all countries, while Steam represents a big player base of almost the entire world. A stronger approach would be a time-series analysis for each country (or maybe even each part of a country that has unique and significant rules or media coverages that might impact the behaviour of players). Either way, it is important to better communicate this. We chose March 2020 as the beginning not to represent the beginning of most country restrictions, but as the beginning of COVID as a major pandemic in in public awareness (which arguably began much earlier for China and around the beginning of March for most of Europe). It was chosen to not mistakenly have March 2020 influence the pre-covid era, when most studies about increased stress began showing effect beginning in March. It’s important to understand that reports and public understanding and consequences are very heterogenic, but March was chosen based on these other studies and the official report of the WHO regarding mental health consideration during the COVID-19 pandemic that was made public (Reference: World Health Organization. (2020). Mental health and psychosocial considerations during the COVID-19 outbreak, 18 March 2020 (No. WHO/2019-nCoV/MentalHealth/2020.1). World Health Organization.) To make this more clearclearer, we added the following paragraph to the manuscript:

“While there is no consent on an exact beginning of the pandemic and countries showing different progressions regarding reported cases and institutional consequences, March 2020 was reported as the 87beginning of stress-related effects in many studies [2–4] and official public consideration of stress-related concerns made by the World Health Organization (WHO; [39]).” (lLines 86-89)

 

Reviewer #2

Thank you so much for this research. Your paper adds, as you say, real-life data to explore theories on the use/impact of serious games. I can't speak to the statistical analysis part.

What went well: Your discussion section answered all the critics I had. For instance; you didn't assume that relaxation games were the solution. People play video games to relax regardless of style or genre; and stress- reducing activities are different for everyone.

Thank you for these kind words. It was important to us to showcase how research and methodological approaches could be used to explain behaviour in various circumstances and why this is a good thing. Data-driven analyses can and should provide new insights about how people act - and always test prevalent theories. We hope we achieved this with this revision.

Questions I still have:

Your findings point out that more people are playing more games and this increased prior to COVID. Now does this coincide with anything else happening in the game environment? Cost? Improved technology? Improved internet access?

This is an interesting point. As a quasi-experimental setup it is almost impossible to evaluate all factors that impact such a comparison and it would be false to assume that March-November 2019 vs. March-November 2020 only had COVID-related differences. In fact, in the revised analysis, we found that an increase in gameplay was expected even without COVID, which is probably related to all factors you’ve mentioned and even more - as interest in gaming seems to increase continuously. Still, we believe that there is meaning in our results, as even with this expected growth, there was a increase in playtime (which could have many possible reasons), but this did not differentiate relaxing from non-relaxing games. We included some of your questions in our revised discussion in the hopes of inspiring more questions around ADPP-influencing factors (see line 264-280).

You ask: Why are people not playing more as a way to decrease stress? Consider that the broader messages are telling us to get up and move, go out for a walk etc. Gaming is part of our 'feel good' activities, but one needs a break from staring at a screen regardless of how stress-reducing it is. 

Your paper is well written, well argued. My two questions fit into how does gaming fit into the broader context. They stem from curiosity more than criticism so don't feel pressured to change anything in the paper.

This is true and an interesting explanation that might need further examination. Our theories suggest that people seek stress-reducing behaviours, especially in times of stress. Games offer these - but there might be a limit. This is a field that is not well understood yet: How well can stress be regulated using video games? How well does a combined effect of outdoor activity and gameplay explain stress reduction, compared to just playing games? And are people aware of this and behave accordingly? This research provides opportunities to ask many more questions, which is exactly what one would hope (even though it lacks universal answers, such is the nature of human behaviour research).

---

## [Decision Letter · Decision Letter 1]

1 Dec 2021

A data-driven approach for examining the demand for relaxation games on Steam during the COVID-19 pandemic

PONE-D-21-07746R1

Dear Dr. Croissant,

We’re pleased to inform you that your manuscript has been judged scientifically suitable for publication and will be formally accepted for publication once it meets all outstanding technical requirements.

Kind regards,

Flávia L. Osório, PhD

Academic Editor

PLOS ONE

Reviewers' comments:

Reviewer's Responses to Questions

**Comments to the Author**

1. If the authors have adequately addressed your comments raised in a previous round of review and you feel that this manuscript is now acceptable for publication, you may indicate that here to bypass the “Comments to the Author” section, enter your conflict of interest statement in the “Confidential to Editor” section, and submit your "Accept" recommendation.

Reviewer #1: All comments have been addressed

Reviewer #2: All comments have been addressed

2. Is the manuscript technically sound, and do the data support the conclusions?

Reviewer #1: (No Response)

Reviewer #2: Yes

3. Has the statistical analysis been performed appropriately and rigorously? 

Reviewer #1: (No Response)

Reviewer #2: Yes

4. Have the authors made all data underlying the findings in their manuscript fully available?

Reviewer #1: (No Response)

Reviewer #2: Yes

5. Is the manuscript presented in an intelligible fashion and written in standard English?

Reviewer #1: (No Response)

Reviewer #2: Yes

6. Review Comments to the Author

Reviewer #1: (No Response)

Reviewer #2: The authors responded to the reviewers comments with thoughtful reflection. They made changes to the paper based upon these recommendations.

7. PLOS authors have the option to publish the peer review history of their article (what does this mean?). If published, this will include your full peer review and any attached files.

Reviewer #1: No

Reviewer #2: No

---

## [Editor Report · Acceptance letter]

7 Dec 2021

PONE-D-21-07746R1 

A data-driven approach for examining the demand for relaxation games on Steam during the COVID-19 pandemic 

Dear Dr. Croissant:

I'm pleased to inform you that your manuscript has been deemed suitable for publication in PLOS ONE. Congratulations! Your manuscript is now with our production department. 

Kind regards, 

on behalf of

Dr. Flávia L. Osório 

Academic Editor

PLOS ONE